# Global trends in smoking-attributable rheumatoid arthritis burden: Insights from GBD 2021

**Kaibin Lin[1]☯, Yang Yi[1]☯, Xi Xu[2]☯, Zheng Wang[1], Yiyue Chen[3,4], Jiafen Liao[4,5], Bing Zhou** [3,4]*

1 School of Computer Science, Hunan First Normal University, Changsha, Hunan, China, 2 School of Computer Science, Hunan University of Technology, ZhuZhou, Hunan, China, 3 Clinical Nursing Teaching and Research Section, The Second Xiangya Hospital of Central South University, Changsha, Hunan, China, 4 Department of Rheumatology and Immunology, The Second Xiangya Hospital of Central South University, Changsha, Hunan, China, 5 Clinical Medical Research Center for Systemic Autoimmune Diseases in Hunan Province, Changsha, Hunan, China

☯ These authors contributed equally to this work.
* zhoubing05@csu.edu.cn

## Abstract

### Background

Smoking is one of the most significant environmental risk factors for Rheumatoid Arthritis (RA). However, there is a lack of research examining the impact of smoking trends on the RA disease burden globally.

### Methods

This study utilized the Global Burden of Disease (GBD) 2021 database to analyze the burden of RA attributable to smoking. Five key indicators were examined: Deaths, Disability-Adjusted Life Years (DALYs), Years Lived with Disability (YLDs), Years of Life Lost (YLLs), and the Socio-Demographic Index (SDI). The analysis was stratified by age, sex, year, and region. Additionally, smoking prevalence and tobacco use data from 2000 to 2021 were extracted from the World Health Organization (WHO) to evaluate trends in smoking and RA burden.

### Results

From 1990 to 2021, while the age-standardized Smoking Attributable Fraction for RA burden metrics generally declined globally, alongside decreasing age-standardized rates (ASR) of smoking-attributable burden in many regions, the absolute global number of both deaths and DALYs due to smoking-attributable RA paradoxically increased (deaths: from 1,792–2,264; DALYs: from 145,727–215,780; all 95% Uncertainty Intervals provided in text). Significant disparities were observed: high-income regions demonstrated greater reductions in smoking-attributable burden than low- and middle-income regions. Males and older populations experienced higher burdens across all metrics. Moderate SDI countries had the highest smoking-attributable

**Data availability statement:** All relevant data are within the manuscript and its Supporting Information files.

**Funding:** This work was supported by the Scientific Research Fund of Hunan Provincial Education Department (grant numbers 23C0427, 23A0643), the Hunan Provincial Natural Science Foundation of China (No.2021JJ40841), and the Research on education and teaching reform of Central South University (No. 2023jy087-3).

**Competing interests:** The authors have declared that no competing interests exist.

age-standardized Deaths and YLLs rate (e.g., Deaths 0.04 per 100,000 population), whereas high SDI countries showed higher YLDs rate (e.g., 3.5 per 100,000 population).

## Conclusions

This study highlights the persistent impact of smoking on the global RA burden and underscores the critical role of tobacco control policies in alleviating this burden. Tailored interventions for high-burden regions (e.g., Eastern Europe and East Asia) and high-risk populations (e.g., middle-aged and older males) are essential. Strengthening early interventions and resource allocation in low- and middle-income regions and enhancing long-term RA management in high-income regions are crucial steps to further reduce the global RA burden.

## 1  Introduction

Rheumatoid arthritis (RA) is a debilitating systemic inflammatory disease [1], with its pathogenesis believed to result from an interaction between genetic predisposition and environmental exposures [2]. Globally, the prevalence of RA is on the rise, with an estimated 17.6 million people affected in 2020, and its age-standardized prevalence rate increasing by 14.1% since 1990 [3]. 40–70% of RA risks are attributed to non-genetic factors, which play an even greater role than genetic predisposition [4], highlighting the importance of studying the role of non-genetic factors in RA's prevalence trends [5]. Among these, smoking has been consistently identified as a significant external risk factor and is closely linked to the pathogenesis of RA [6–9]. Specifically, smoking is thought to promote RA development by inducing citrullination of proteins, particularly in the lungs of genetically susceptible individuals, which can lead to the generation of anti-citrullinated protein antibodies (ACPAs), key autoantibodies in RA [9].

Smoking is recognized as the most preventable cause of chronic non-communicable diseases (NCDs) and premature mortality worldwide [10]. In 2003, the World Health Organization (WHO) launched the Framework Convention on Tobacco Control (FCTC), aiming to create a smoke-free world [11]. This aligns with the United Nations' Sustainable Development Goal of reducing premature deaths from NCDs by one-third by 2030 [12]. Tobacco control is thus one of the most effective strategies for reducing the burden of NCDs [13]. Regular assessments of the effectiveness and implementation of tobacco control measures are therefore essential.

However, while the impact of smoking on RA is well-documented and the FCTC represents a global commitment to tobacco control, there is a paucity of comprehensive global analyses examining the long-term trends in smoking-attributable RA burden specifically since the FCTC's implementation. Understanding these trends is crucial for evaluating the real-world impact of tobacco control efforts on specific NCDs like RA and for guiding future public health strategies.

This study aims to assess global trends in the RA burden attributable to smoking since the implementation of the WHO FCTC, using data from GBD 2021 and WHO tobacco prevalence reports. Findings will inform health policy and tobacco control strategies aimed at reducing the RA burden.

## 2 Methods

### 2.1 Data sources and definitions

The data analyzed in this study regarding the RA burden attributable to smoking were obtained from the GBD 2021. The GBD 2021 provides comprehensive and comparable annual epidemiological estimates for 371 diseases and injuries across 204 countries and territories and 21 GBD super-regions from 1990 to 2021. All GBD data are publicly accessible through the Global Health Data Exchange (GHDx) (https://ghdx.healthdata.org/gbd-2021/sources) [14]. Detailed information on GBD methodologies, data inputs, and statistical modeling can be found in previous GBD capstone publications [15]. In GBD 2021, causes are hierarchically categorized; RA is classified as a Level 3 cause under musculoskeletal disorders [16]. The case definition for RA in GBD 2021 was based on the 1987 revised criteria of the American College of Rheumatology [17].

From the GBD 2021 database, for RA attributable to smoking, we extracted both the absolute numbers and the age-standardized rates (ASR, expressed per 100,000 population) for the following key burden metrics: Deaths, Disability-Adjusted Life Years (DALYs), Years Lived with Disability (YLDs), and Years of Life Lost (YLLs). These data were obtained stratified by age group (standard GBD 5-year age groups), sex (male, female, and both), location (country/territory and GBD super-region), and year (annually from 1990 to 2021). Additionally, Socio-Demographic Index (SDI) values were extracted for each location and year from GBD 2021. Our study aimed to include all locations for which GBD 2021 provides these estimates.

For trends in tobacco exposure, we collected regional and country-level data on the age-standardized prevalence of current tobacco smoking (%) and the age-standardized prevalence of current tobacco use (%) from the World Health Organization (WHO) Global Health Observatory (GHO) data repository. These prevalence estimates typically refer to persons aged 15 years and older, where specified by WHO. Data were extracted for the specific years 2000, 2005, 2007, 2010, 2015, 2018, 2020, and 2021 (to align our analysis with the GBD 2021 end year). These data were primarily sourced under the indicator tracking SDG Target 3.a: "Strengthen the implementation of the World Health Organization Framework Convention on Tobacco Control in all countries" [18]. Further details on global smoking prevalence can be found in WHO's "Global Tobacco Epidemic Report" series [19].

### 2.2 Data integrity and visualization mapping

This study utilized the final, publicly available, and pre-processed estimates from both GBD 2021 and WHO, which have undergone extensive internal processing by their respective organizations to ensure data quality and completeness. Therefore, our study team did not perform additional data cleaning or imputation. For map visualizations (e.g., Fig 1), country-level data were merged with a Natural Earth shapefile; a predefined country name mapping was applied to reconcile naming discrepancies and maximize data display. Regions depicted as "No Data" on maps reflect either the absence of data for that indicator in our source files or non-representation in the 1:110m shapefile.

### 2.3 ASR, EAPC and percentage change

All analyses of trends in RA burden attributable to smoking, including the calculation of ASR, estimated annual percentage changes (EAPC), and overall percentage changes, were conducted and presented stratified by GBD super-region, sex (male and female), and relevant age groups, in addition to analyses by SDI quintiles. This stratified approach was employed to investigate disparities in the burden across these demographic and developmental subgroups.

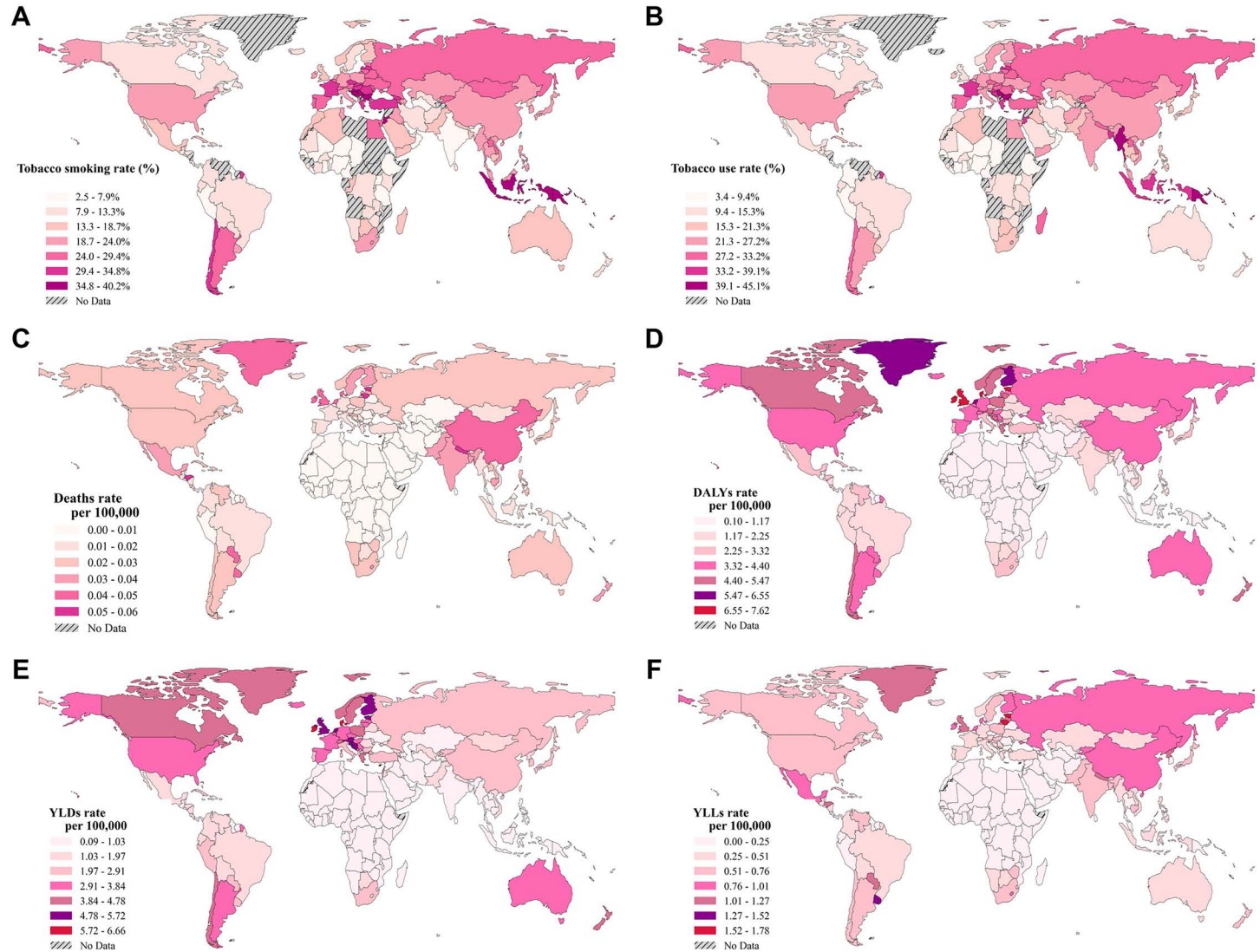

**Fig 1. Burden of RA attributable to smoking in 2021 across 204 countries and territories.** Map data obtained from Natural Earth (public domain).

We calculated the ASR per 100,000 population using Equation (1) [20]. ASR is a measure designed to eliminate the influence of differences in population age structures to the greatest extent possible [21]. All cases and their corresponding ASRs per 100,000 population were reported with 95% uncertainty intervals (UIs).

$$ASR = \frac{\sum_{i=1}^{n} \alpha_i W_i}{\sum_{i=1}^{n} W_i}$$

(1)

Here, $\alpha_i$ represents the age-specific rate in the $i$th age group, $W_i$ denotes the number of individuals in the corresponding age group within the GBD 2021 standard population, and $n$ is the total number of age groups. Additionally, this study used percentage change to represent the trend of the RA burden attributable to smoking in 2021 compared to 1990.

$$Percentage\ Change = \frac{APAS_{2021} - APAS_{1990}}{APAS_{1990}} \times 100\%$$

(2)

Here, APAS is Age-standardised Percentage Attributable to Smoking.

The EAPC is widely used to track trends in indicators such as prevalence and incidence over specific time periods [22]. We calculated the EAPC of ASR to assess the average trend of change over a given time interval. The calculation of EAPC is based on fitting a regression model to the natural logarithm of the ASR, using time as the independent variable. Each observation's natural logarithm is fitted to a straight line, and the slope of this line is used to compute the EAPC [23]:

$$\ln(ASR) = \alpha + \beta x + \gamma \tag{3}$$

$$EAPC_{with\ 95\%\ CI} = 100 \times \left(e^{\beta} - 1\right) \tag{4}$$

Here, $x$ represents the year, $y$ denotes the natural logarithm of rates, $\alpha$ is the intercept, $\beta$ is the slope, and $\epsilon$ represents the random error. If the EAPC and its 95% confidence interval (CI) include zero, we consider the change in ASR to be statistically insignificant. Conversely, if the EAPC and its 95% CI are entirely above or below zero, we conclude that the ASR demonstrates an increasing or decreasing trend over time, respectively [24]. Data preprocessing and mathematical computations were conducted using Python's numpy library, while the statsmodels library was employed to fit a linear regression model for calculating the slope and EAPC. The EAPC calculation, by regressing the natural logarithm of ASR against calendar year, inherently assumes an approximately log-linear trend over the evaluated period. While formal statistical testing of all regression assumptions (e.g., linearity of log-transformed ASR, homoscedasticity, normality of residuals) was not performed for every EAPC estimated due to the large number of trends analyzed across various stratifications, we visually inspected scatter plots of ln(ASR) versus year for major global and regional trends. These qualitative assessments generally indicated that a log-linear model provided a reasonable approximation for summarizing the average annual percentage change over the 32-year study period. The EAPC is utilized in this study as a standardized descriptive metric to quantify and compare the overall direction and magnitude of secular trends in ASR, a common practice in GBD analyses and epidemiological reporting [22,23].

## 2.4 Deaths, DALYs, YLDs, and YLLs

Deaths refer to the proportion of disease-related deaths per 100,000 population. Age-standardized Deaths are used to account for differences in Deaths rate caused by variations in population size and age structure across different regions [25]. DALYs are the standard metric for quantifying disease burden, representing the years of healthy life lost due to disease from onset to death. DALYs are calculated as the sum of YLLs and YLDs, and the formula is as follows:

$$DALYs = YLLs + YLDs \tag{5}$$

Here, YLLs are calculated as the product of the estimated number of age-sex-location and year-specific deaths and the standard expected years of life remaining at the age of death due to a specific cause. YLDs are computed by multiplying the prevalence of specific disease sequelae by the corresponding disability weights [14].

## 2.5 SDI

Over the past three decades, socio-demographic development has been a major contributor to health improvements [20]. The SDI is a composite indicator representing the geometric mean of three parameters: lag-distributed income per capita, average years of schooling, and fertility rate among women under 25 years of age in a given region. For GBD 2021, the final SDI values were multiplied by 100 for reporting purposes. An SDI of 0 indicates the theoretical minimum level of development related to health (lowest income and years of schooling, highest fertility rate), while an SDI of 100 represents the theoretical maximum level (highest income and years of schooling, lowest fertility rate) [14].

This study was based on publicly available, aggregated, and de-identified secondary data obtained from the GBD 2021 study and WHO public data repositories. As such, formal ethical review approval and informed consent from participants were not required for this secondary data analysis.

## 3 Results

### 3.1 Trends in the global and regional RA burden Attributable to smoking from 1990 to 2021

From 1990 to 2021, the age-standardized Smoking Attributable Fraction for RA burden metrics generally declined across GBD super-regions, although the magnitude and direction of this change varied substantially (Table 1).

Regarding age-standardized Deaths rate attributable to smoking, the African region demonstrated the most substantial percentage reduction, decreasing by 50.94% (EAPC = −2.54, 95% CI: −2.65 to −2.42) from 1990 to 2021. In contrast, the Western Pacific region exhibited the smallest decline at 7.73% (EAPC = −0.14, 95% CI: −0.21 to −0.07) over the same period.

**Table 1. Age-standardised rates (per 100,000 population) of rheumatoid arthritis burden attributable to smoking in 1990 and 2021, and trends from 1990 to 2021, by GBD Super-Region.**

| | Age-standardised Percentage Attributable to Smoking | | Percentage change 1990–2021(%) (95% UI) | EAPC 1990–2021(%) (95% CI) |
|---|---|---|---|---|
| | 1990(%) (95% UI) | 2021(%) (95% UI) | | |
| **Deaths** | | | | |
| Africa | 6.44 (4.01,8.51) | 3.16 (1.84,4.22) | −50.94 (−54.04,−50.44) | −2.54 (−2.65,−2.42) |
| Americas | 8.48 (6.68,10.35) | 4.51 (3.00,5.90) | −46.83 (−55.11,−42.93) | −1.84 (−1.88,−1.80) |
| Eastern Mediterranean | 5.04 (3.26,7.08) | 3.17 (2.07,4.48) | −37.08 (−36.67,−36.52) | −1.62 (−1.70,−1.53) |
| Europe | 8.63 (6.82,10.42) | 4.99 (3.82,6.11) | −42.18 (−43.97,−41.42) | −0.89 (−0.94,−0.84) |
| South-East Asia | 5.74 (3.88,7.61) | 6.30 (4.97,7.70) | 9.82 (1.23, 28.13) | −0.79 (−0.83,−0.75) |
| Western Pacific | 8.47 (6.26,10.84) | 7.82 (5.36,10.25) | −7.73 (−14.27,−5.44) | −0.14 (−0.21,−0.07) |
| **DALYs** | | | | |
| Africa | 6.22 (4.86,7.72) | 6.52 (5.11,7.96) | 4.92 (3.06,5.19) | −1.84 (−1.95,−1.73) |
| Americas | 10.97 (8.64,13.28) | 4.15 (3.18,5.22) | −62.12 (−63.17,−60.65) | −1.78 (−1.81,−1.74) |
| Eastern Mediterranean | 6.23 (4.85,7.76) | 9.35 (7.33,11.25) | 50.07 (44.93,51.14) | −1.30 (−1.36,−1.25) |
| Europe | 11.38 (9.08,13.63) | 4.27 (3.34,5.35) | −62.46 (−63.28,−60.78) | −0.54 (−0.60,−0.48) |
| South-East Asia | 5.82 (4.50,7.20) | 8.29 (6.57,9.98) | 42.42 (38.66,45.99) | −1.12 (−1.16,−1.09) |
| Western Pacific | 9.45 (7.48,11.46) | 3.62 (2.79,4.52) | −61.69 (−62.69,−60.61) | −0.41 (−0.47,−0.36) |
| **YLDs** | | | | |
| Africa | 5.84 (4.63,7.22) | 4.62 (3.60,5.69) | −20.83 (−22.36,−21.16) | −1.58 (−1.69,−1.46) |
| Americas | 11.50 (9.10,13.89) | 3.86 (3.03,4.77) | −66.45 (−66.69,−65.71) | −1.81 (−1.84,−1.77) |
| Eastern Mediterranean | 6.69 (5.27,8.14) | 3.59 (2.81,4.39) | −46.44 (−46.65,−46.06) | −1.26 (−1.32,−1.20) |
| Europe | 12.07 (9.53,14.44) | 9.69 (7.58,11.64) | −19.68 (−20.43,−19.41) | −0.64 (−0.69,−0.59) |
| South-East Asia | 5.64 (4.46,6.89) | 8.12 (6.42,9.74) | 44.00 (41.30,43.97) | −1.29 (−1.34,−1.25) |
| Western Pacific | 9.63 (7.74,11.50) | 6.78 (5.28,8.27) | −29.62 (−31.87,−28.04) | −0.58 (−0.65,−0.50) |
| **YLLs** | | | | |
| Africa | 7.57 (4.47,9.79) | 8.82 (5.95,11.44) | 16.55 (16.93,32.95) | −2.43 (−2.56,−2.31) |
| Americas | 9.72 (7.65,11.78) | 5.60 (4.33,6.85) | −42.43 (−43.45,−41.81) | −1.88 (−1.92,−1.85) |
| Eastern Mediterranean | 5.41 (3.49,7.37) | 7.81 (6.22,9.42) | 44.34 (27.76,78.31) | −1.58 (−1.69,−1.48) |
| Europe | 9.92 (7.88,11.94) | 4.76 (3.11,6.21) | −51.96 (−60.52,−47.97) | −0.59 (−0.66,−0.52) |
| South-East Asia | 6.03 (4.03,7.91) | 3.48 (2.23,4.78) | −42.31 (−44.58,−39.53) | −0.77 (−0.80,−0.75) |
| Western Pacific | 9.11 (6.64,11.77) | 3.87 (2.14,5.09) | −57.52 (−67.76,−56.76) | −0.02 (−0.05,−0.10) |

For age-standardized Disability-Adjusted Life Years (DALYs) rate, the Americas experienced the largest decrease (−62.12%, EAPC = −1.78, 95% CI: −1.81 to −1.74), while the European region showed a more modest reduction (−62.12%, EAPC = −0.54, 95% CI: −0.60 to −0.48). A similar pattern was observed for age-standardized Years Lived with Disability (YLDs) rate, with the Americas again showing the most significant improvement (−66.45%, EAPC = −1.81, 95% CI: −1.84 to −1.77), and the Western Pacific the least (−29.62%, EAPC = −0.58, 95% CI: −0.65 to −0.50).

Finally, age-standardized Years of Life Lost (YLLs) rate decreased most markedly in the African region (−16.55%, EAPC = −2.43, 95% CI: −2.56 to −2.31), with the smallest reduction seen in the Western Pacific region (−57.52%, EAPC = −0.02, 95% CI: −0.05 to −0.10).

These varied regional trends likely reflect differences in the implementation and effectiveness of tobacco control policies, healthcare access and quality, and prevailing cultural practices related to smoking.

In 2021, significant regional disparities were evident in both tobacco exposure and the RA burden attributable to smoking (Fig 1).

Regions with the highest tobacco smoking rate (%) (Fig 1A) and tobacco use rate (%) (Fig 1B) included Eastern Europe, East Asia (e.g., China), Southeast Asia (e.g., Indonesia), parts of South America (e.g., Argentina and Uruguay), and North Africa. For example, in some Eastern European countries, tobacco smoking rates reached approximately 40%, while in sub-Saharan Africa, they were below 10%.

Correspondingly, the highest smoking-attributable age-standardized Deaths rate (Fig 1C) and DALYs rate (Fig 1D) per 100,000 population were observed in Eastern Europe, East Asia (e.g., China), and parts of South America. In these high-burden regions, age-standardized Deaths rates exceeded 0.05 per 100,000 population, while age-standardized DALYs rate reached as high as 6 per 100,000 population. In contrast, sub-Saharan Africa, Western Europe, and the Middle East reported lower burdens for these metrics, with age-standardized Deaths rate below 0.01 per 100,000 population in these areas.

The distribution of age-standardized YLDs rate (Fig 1E) and age-standardized YLLs rates (Fig 1F) per 100,000 population also showed distinct regional patterns in 2021. High-income countries (e.g., North America, Eastern Europe, and Western Europe) generally exhibited the highest YLDs rates, potentially reflecting longer lifespans and more prolonged management of chronic conditions. For instance, in North America, age-standardized YLDs rates reached approximately 5 per 100,000 population, compared to less than 1 per 100,000 population in sub-Saharan Africa.

Regions with the highest age-standardized YLLs rate, indicating significant premature mortality attributable to smoking-related RA, were Eastern Europe, East Asia, and parts of South America, where rates were approximately 1.2 per 100,000 population. In contrast, low-burden regions such as Western Europe and sub-Saharan Africa had age-standardized YLLs rate below 0.5 per 100,000 population.

## 3.2 Temporal and regional trends in smoking-attributable RA burden

Between 2000 and 2021, global tobacco smoking rate and tobacco use rates showed a significant decline. Smoking prevalence decreased consistently across all regions (Fig 2A), with high-income regions showing the most pronounced reduction, dropping from approximately 35% in 2000 to about 15% in 2021. In contrast, low and middle-income regions experienced smaller reductions, with tobacco smoking rate decreasing from around 25% in 2000 to about 20% in 2021. The trends in tobacco use rates mirrored those of tobacco smoking rate (Fig 2B), with high-income regions exhibiting the fastest decline, from approximately 45% in 2000 to about 20% in 2021. Conversely, low-income regions showed limited improvement, with rates remaining around 35%. These patterns can be attributed to strict public health policies in high-income regions, such as smoking bans and health education campaigns, while weaker tobacco control measures and cultural influences likely contributed to the slower or stagnant decline in low- and middle-income regions.

Indicators of the smoking-attributable RA burden, including Deaths, DALYs, YLDs, and YLLs, also showed a downward trend during this period (Figs 2C-F). Among these, the reductions in Deaths and YLLs were the most significant (Figs 2C and 2F). Deaths in high-income regions dropped from approximately 0.05 per 100,000 population in 2000 to about 0.02

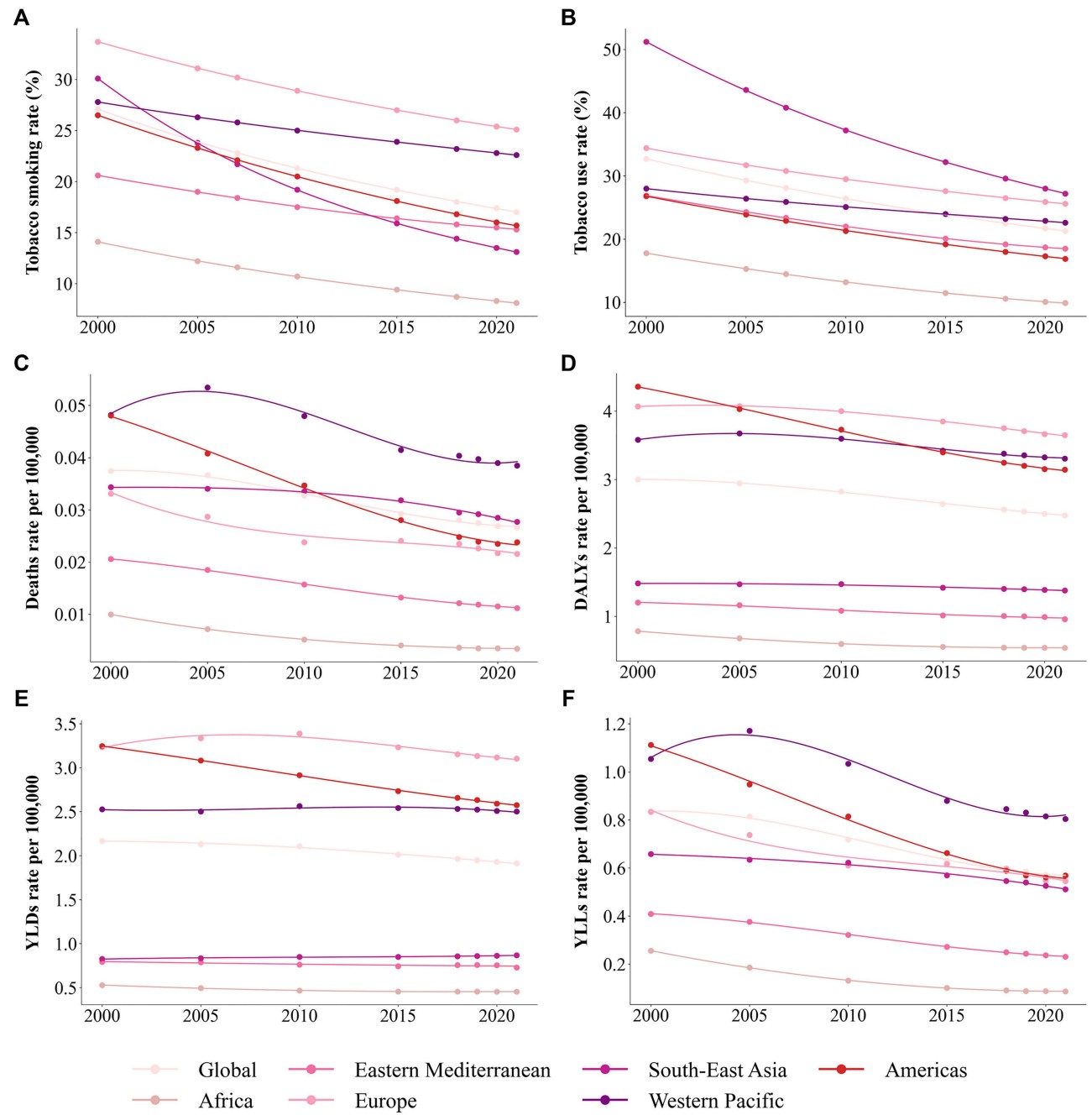

**Fig 2. Smoking-Attributable RA Burden from 2000 to 2021: Global Temporal Trends and Regional Differences in Smoking Prevalence and Burden Indicators.**

per 100,000 population in 2021, while in low-income regions, they declined from around 0.03 per 100,000 population in 2000 to about 0.02 per 100,000 population in 2021. YLLs rates showed a more pronounced decrease in high-income regions, falling from about 1.2 per 100,000 population in 2000 to approximately 0.6 per 100,000 population in 2021. By contrast, low and middle-income regions experienced slower declines.

## 3.3 Age and sex differences in smoking-attributable RA burden

In 2021, the burden of RA attributable to smoking exhibited significant differences across age, sex, and regions (Figs 3 and 4). From an age and sex perspective, Deaths, DALYs, YLDs, and YLLs increased with age for both males and females (Fig 3). Deaths (Fig 3A) increased significantly in individuals aged 85 and above, with rates of approximately 0.25 per 100,000 population for males and 0.15 per 100,000 population for females. Deathss were higher in males across all age groups, with the disparity becoming more pronounced in older age groups. DALYs (Fig 3B) peaked between the ages of 50 and 80 and then gradually declined, though they remained high in individuals aged 85 and above. Among individuals aged 70–74, DALYs were 4.5 per 100,000 population for males and 3.0 per 100,000 population for females. YLDs (Fig 3C) indicated the heaviest disability burden in middle-aged and older populations (50–74 years). In the 65–69 age

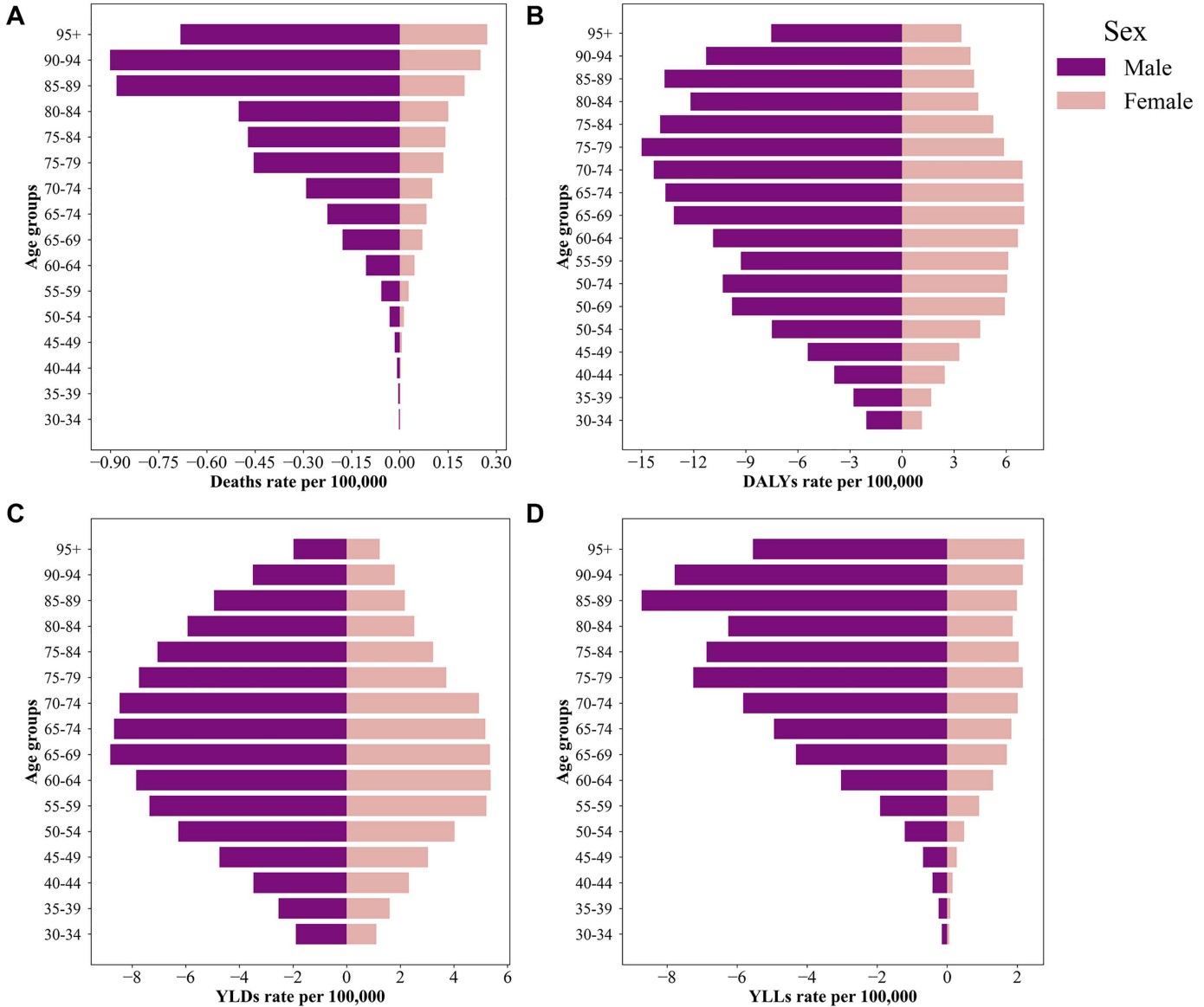

**Fig 3. Sex and age differences in smoking-attributable RA burden in 2021.**

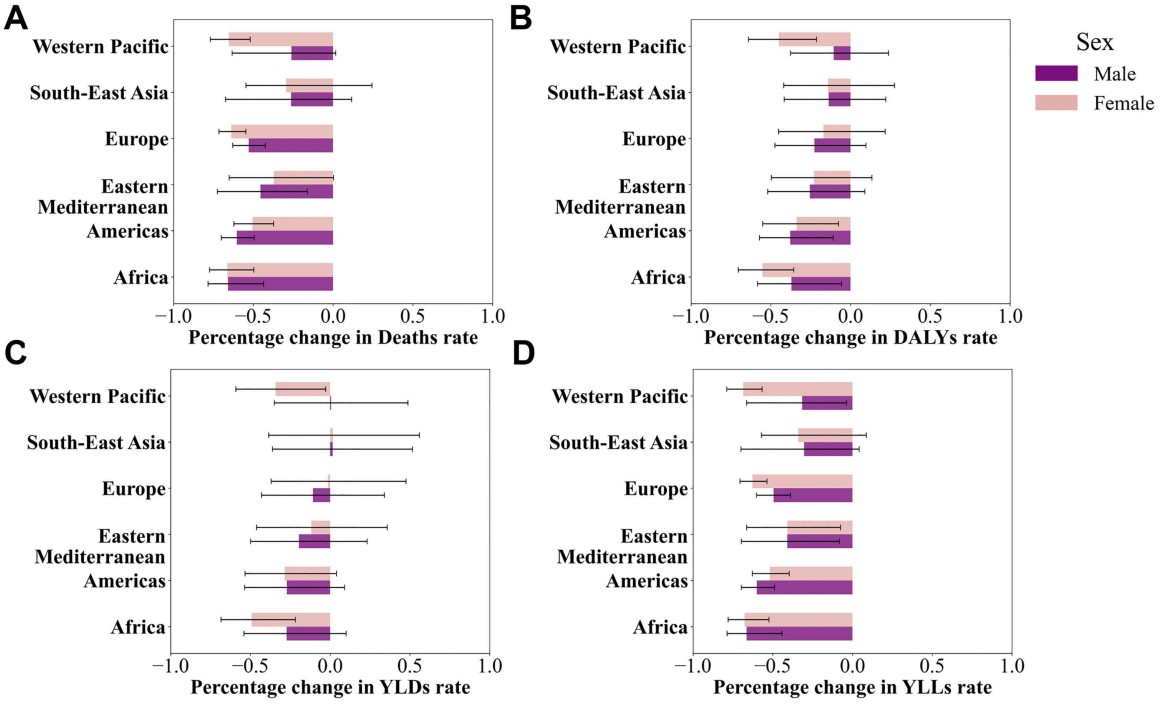

**Fig 4. Sex differences and percentage change trends in smoking-attributable RA burden from 1990 to 2021.**

group, YLDs rates were 3.5 per 100,000 population for males and 2.5 per 100,000 population for females, likely reflecting disease progression and functional impairment associated with RA. YLLs (Fig 3D) showed that premature mortality due to smoking had a greater impact on males. Among individuals aged 75 and above, YLLs were 1.2 per 100,000 population for males and 0.6 per 100,000 population for females.

From a regional perspective, the burden trends between 2000 and 2021 varied significantly across regions and sexes (Fig 4). Deaths (Fig 4A) showed the largest decline among males in the Americas, with an EAPC of −1.88%, while the smallest reduction was observed among females in the Western Pacific (−0.14%). DALYs (Fig 4B) followed a similar pattern, with the largest reduction among males in the Americas (EAPC=−1.78%) and smaller improvements among females in Europe and the Western Pacific. YLDs (Fig 4C) indicated the most significant reduction in disability burden among males in the Americas (−1.81%), whereas the smallest change was observed among females in the Western Pacific (approximately −0.58%). YLLs (Fig 4D) revealed the most notable improvement in years of life lost among males in the Americas and Africa (EAPC=−2.43% and −1.88%, respectively), while females in Southeast Asia and the Western Pacific showed smaller changes.

### 3.4 Relationship between smoking-attributable RA burden and SDI

The Socio-Demographic Index (SDI) plays a significant role in modulating the burden of RA attributable to smoking (Fig 5). Both Deaths (Fig 5A) and YLLs (Fig 5D) exhibit an "inverted U-shaped" trend, peaking in countries with moderate SDI levels (SDI around 0.5–0.6). For instance, Deaths in moderate SDI countries were the highest, approximately 0.04 per 100,000 population, while YLLs reached around 1.2 per 100,000 population. In contrast, countries with low and high SDI levels experienced lower burdens (both below 0.5 per 100,000 population). This trend highlights the disproportionately severe burden of smoking-related Deaths in moderate SDI countries.

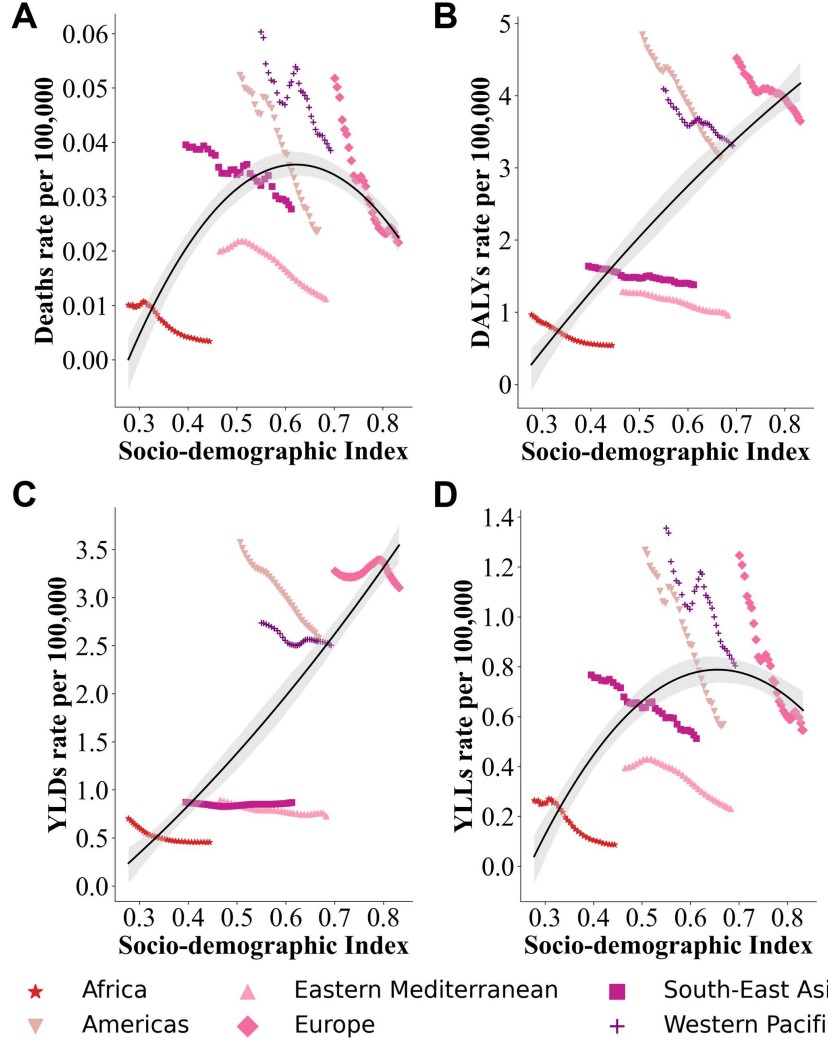

**Fig 5. Relationship between SDI and smoking-attributable RA burden.**

DALYs (Fig 5B) and YLDs (Fig 5C), however, show a positive correlation with increasing SDI. In high SDI countries, the DALYs burden reached approximately 4 per 100,000 population, while YLDs were around 3.5 per 100,000 population, significantly higher than in low SDI countries (both DALYs and YLDs below 1 per 100,000 population). This indicates that although high SDI countries experience a lighter burden of smoking-related Deaths, the prolonged survival of patients leads to a higher disability-related burden.

## 4 Discussion

Rheumatoid arthritis (RA) remains a critical global health issue [5]. Over the past three decades, while the global age-standardized prevalence and incidence rate of RA have reportedly increased [3], our study found that the age-standardized rate of smoking-attributable RA Deaths and Disability-Adjusted Life Years (DALYs) showed a significant decline globally from 1990 to 2021. This decline in age-standardized smoking-attributable burden likely reflects, in part, the impact of global tobacco control efforts leading to reduced smoking prevalence over time, alongside advancements

in RA treatment paradigms and overall improvements in disease management which may reduce the severity or mortality of RA [26]. However, despite these encouraging trends in age-standardized rates, the absolute number of Deaths from smoking-attributable RA increased from approximately 1,792 (95% UI: 1,323.15 to 2,227.5) in 1990–2,264 (95% UI: 1,509.36 to 2,911.43) in 2021, while the absolute number of DALYs rose from approximately 145,727 (95% UI: 102,543.07 to 200,899.79) to 215,780 (95% UI: 147,152.90 to 300,761.09) during the same period. The age-standardized proportions of Deaths and DALYs attributable to smoking in 2021 were 5.93% (95% UI: 4.33% to 7.43%) and 6.9% (95% UI: 5.45% to 8.33%), respectively, underscoring the persistent and substantial impact of smoking on the RA burden.

Several factors likely contribute to this divergence where age-standardized rate of smoking-attributable burden decrease while absolute numbers rise, a phenomenon also observed in the context of overall declining smoking rate not immediately or proportionally translating into sharp reductions in all smoking-related disease burdens. Firstly, there is a well-documented lag effect between peak smoking exposure and the subsequent development and clinical manifestation of smoking-related diseases, including RA. The pathophysiological processes triggered by tobacco smoke that contribute to autoimmunity and RA pathogenesis often take many years, or even decades, to culminate in a diagnosable condition [27]. Thus, a significant portion of the current smoking-attributable RA burden reflects the consequences of higher smoking prevalence in previous decades. Secondly, global demographic transitions, notably population growth and population aging, are major drivers. Population growth inherently increases the number of individuals at risk, and even with declining rates, a larger population base can lead to a higher absolute number of cases and deaths [28]. More specifically, as global life expectancy rises, the proportion of elderly individuals in the population increases; older age is a strong risk factor for RA, and older individuals may have had longer cumulative exposure to smoking or experience progression of RA potentially initiated by smoking earlier in life [29]. Thirdly, advancements in RA management, while beneficial for patients, may lead to increased survival durations with the condition, thereby potentially increasing the duration of disability (Years Lived with Disability - YLDs) and contributing to the absolute DALYs burden if smoking was an etiological factor [30].

This study systematically assessed global trends and regional disparities in smoking-attributable RA burden from 1990 to 2021. Our results indicate a significant reduction in the global age-standardized smoking-related RA burden over the past 32 years, although substantial differences persist across regions and sexes.

High-income regions generally showed the most significant improvements in reducing this burden, which can likely be attributed to more established and stringent tobacco control policies alongside advanced healthcare systems capable of better RA management and smoking cessation support. Conversely, many low- and middle-income regions exhibited slower progress or even increases in certain absolute burden metrics, highlighting the urgent need for enhanced public health interventions, stricter implementation of the WHO Framework Convention on Tobacco Control (FCTC), and culturally tailored tobacco control strategies in these regions. These findings are consistent with trends observed in previous RA studies and reports on global tobacco control [31,32].

From a sex and age perspective, while global RA prevalence is typically higher among females than males [3], our findings indicate that in the context of smoking-attributable RA burden, males exhibited higher rates across all indicators, with the greatest disparities observed in Deaths and Years of Life Lost (YLLs). This reflects the historically higher prevalence and intensity of smoking among males in many parts of the world, leading to a greater smoking-related health burden, a pattern consistent with prior research on smoking and RA pathogenesis, as well as overall smoking-attributable disease burden [5,9,33].

The highest levels of smoking-attributable DALYs and Deaths were observed in older populations. Specifically, Deaths increased significantly in individuals aged 85 and above, while DALYs peaked between the ages of 50 and 80. This concentration of burden in older adults may be influenced by the cumulative effects of long-term smoking, the natural history of RA progression, and the presence of comorbid musculoskeletal conditions such as osteoporosis and osteoarthritis, as well as other systemic diseases common in the elderly [34]. The clinical implication of this is significant, underscoring the need for integrated care approaches for older RA patients, particularly those with a history of smoking. Such approaches

should combine RA management with proactive screening and management of smoking-related comorbidities and continued support for smoking cessation, even in later life, to mitigate further health deterioration [35]. Tobacco control efforts should therefore particularly target middle-aged males to prevent future burden, while health systems need to strengthen RA treatment and comprehensive health management, including comorbidity care, in older populations.

Disparities in burden improvements between regions were notable. Males in some high-income regions showed substantial improvements, while females in certain low- and middle-income regions experienced relatively limited progress. These findings further underscore the need for more targeted tobacco control policies and RA health management strategies that are sensitive to both regional context and sex-specific smoking patterns and healthcare access.

In terms of Socio-Demographic Index (SDI)-related differences, moderate SDI countries experienced the highest age-standardized smoking-attributable Deaths and YLLs rate, while high SDI countries, despite lower smoking-attributable mortality likely due to better healthcare resources and disease management, showed significant YLDs and DALYs rate, reflecting prolonged patient survival with disability. In contrast, low SDI countries exhibited overall lower reported burdens for some indicators; however, this might be an underestimation. The availability and quality of health data, including cause-of-death registration and RA diagnosis capabilities, are often more limited in low SDI settings, and economic constraints can impede access to care and thus affect disease reporting and burden estimation [36,37]. Therefore, the true burden in these regions could be higher than GBD estimates suggest, emphasizing the need for strengthening health information systems globally. For example, infrequent health surveys or incomplete civil registration and vital statistics (CRVS) systems in some low-SDI countries can lead to sparser primary data inputs for both RA occurrence and risk factor prevalence, which GBD models attempt to address but may still result in wider uncertainty or potential bias in estimates for these regions.

Despite shedding light on the long-term impact of smoking on RA burden, this study has several limitations. First, as acknowledged by the GBD collaboration, data from some low-income countries may be sparse or of lower quality, potentially leading to wider uncertainty in estimates or underestimation of the actual burden, even with sophisticated modeling techniques. This inherent limitation means that while GBD provides the best available estimates, the precision and accuracy for some low-SDI countries might be lower compared to high-SDI countries with more robust health information infrastructure. Consequently, comparisons involving these regions should be interpreted with this context in mind. Second, our analysis of WHO tobacco data relied on prevalence estimates for discrete years, which may not fully capture nuanced year-to-year fluctuations in smoking behavior. Third, this study did not address interactions between smoking exposure characteristics (e.g., intensity, duration, type of tobacco product) and other environmental or genetic risk factors for RA, which may limit a deeper understanding of the mechanisms driving the observed burden. Furthermore, while the GBD methodology incorporates robust uncertainty analysis, providing 95% Uncertainty Intervals for all estimates which we have reported, this study did not perform additional study-specific sensitivity analyses, such as exploring the impact of alternative model assumptions or varying definitions for smoking exposure beyond what is provided by GBD. Future research could explore these aspects to further assess the robustness of specific regional or temporal findings.

## 5 Conclusions

This study systematically assessed global trends in the burden of RA attributable to smoking from 1990 to 2021, as well as differences by sex, age, and region. While the global burden of smoking-related RA has shown a declining trend, significant disparities persist across regions and between sexes. The findings highlight the critical role of tobacco control policies in reducing the RA burden. Targeted interventions for high-burden regions (e.g., Eastern Europe and East Asia) and high-risk populations (e.g., middle-aged and older males) are essential. Furthermore, strengthening long-term RA management in high-income countries and improving early intervention and resource allocation in low and middle-income countries are imperative to further alleviate the global RA burden.

## Supporting information

**S1 Data. Minimal dataset for the study.** This compressed ZIP archive contains all underlying data required to reproduce the figures and tables presented in the manuscript.
(ZIP)

## Author contributions

**Conceptualization:** Kaibin Lin.

**Data curation:** Yang Yi.

**Formal analysis:** Xi Xu.

**Investigation:** Yiyue Chen.

**Methodology:** Zheng Wang.

**Writing – original draft:** Jiafen Liao, Bing Zhou.

**Writing – review & editing:** Bing Zhou.

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
