## [Decision Letter · Decision Letter 0]

8 May 2025

PONE-D-25-01779Global Trends in Smoking-Attributable Rheumatoid Arthritis Burden: Insights from GBD 2021PLOS ONE

Dear Dr. Zhou,

Thank you for submitting your manuscript to PLOS ONE. After careful consideration, we feel that it has merit but does not fully meet PLOS ONE’s publication criteria as it currently stands. Therefore, we invite you to submit a revised version of the manuscript that addresses the points raised during the review process. Please submit your revised manuscript by Jun 22 2025 11:59PM. If you will need more time than this to complete your revisions, please reply to this message or contact the journal office at plosone@plos.org . Please include the following items when submitting your revised manuscript:

We look forward to receiving your revised manuscript.

Kind regards,

Paul Obeng, MEd, MSc., M.Phil.

Academic Editor

PLOS ONE

Journal Requirements:

2. We note that your Data Availability Statement is currently as follows: [All relevant data are within the manuscript and its Supporting Information files]

“This work was supported by Scientific Research Fund of Hunan Provincial Education Department(grant number 23C0427, 23A0643), Hunan Provincial Natural Science Foundation of China (No.2021JJ40841), Research on education and teaching reform of Central South University (No. 2023jy087-3).”

6. We note that Figure 1 in your submission contain [map/satellite] images which may be copyrighted. All PLOS content is published under the Creative Commons Attribution License (CC BY 4.0), which means that the manuscript, images, and Supporting Information files will be freely available online, and any third party is permitted to access, download, copy, distribute, and use these materials in any way, even commercially, with proper attribution. For these reasons, we cannot publish previously copyrighted maps or satellite images created using proprietary data, such as Google software (Google Maps, Street View, and Earth). For more information, see our copyright guidelines: http://journals.plos.org/plosone/s/licenses-and-copyright.

Additional Editor Comments:

**Introduction**

While the authors mention that RA prevalence is rising globally, quantitative data or regional variability would enhance context. Consider including global or regional prevalence/incidence rates or burden metrics (e.g., DALYs or mortality). For example, in 2019, RA affected approximately X million people globally, contributing to Y DALYs and Z deaths (cite GBD source)."Insufficient explanation of smoking’s pathophysiological role. Although smoking is cited as a major external risk factor, the introduction could briefly explain the mechanism by which smoking contributes to RA pathogenesis to help justify its inclusion. Add one or two sentences on how smoking induces citrullination, promotes autoantibody production, or triggers immune dysregulation relevant to RA.Need to define the research gap more explicitly. The introduction could better articulate what specific gaps this study addresses. For example, is there a lack of global time-trend analyses? Have previous studies not linked smoking and RA in the context of FCTC implementation?The final paragraph outlines the aim, but the sentence is lengthy and could be more concise and focused. For example, the objectives could be modified to “This study aims to assess global trends in the RA burden attributable to smoking since the implementation of the WHO FCTC, using data from GBD 2021 and WHO tobacco prevalence reports. Findings will inform health policy and tobacco control strategies aimed at reducing the RA burden.”"Interestingly, 40–70% of RA risks are attributed to non-genetic factors..." Consider removing “Interestingly,” as it adds unnecessary subjectivity.Consider specifying whether the study focuses on incidence, prevalence, mortality, or DALYs for RA to avoid ambiguity.

**Method **

While the sources of data are cited, there is insufficient detail on:

Which specific variables were extracted from GBD 2021 and WHO datasets.How countries or regions were selected, especially with reference to completeness of data over time.Handling of missing or inconsistent data, if any. Include a description of inclusion/exclusion criteria, any data cleaning steps, and how missing data were addressed, if applicable.

The selection of years for WHO tobacco prevalence data (2000, 2005, 2007, etc.) appears arbitrary. Clarify the rationale behind choosing these years and explain whether they align with WHO data availability, FCTC milestones, or analytical objectives.The linear regression model used for EAPC assumes a log-linear trend across time, but this assumption is not discussed or evaluated. A brief note on whether assumptions of linearity, homoscedasticity, and normality of residuals were checked would strengthen the rigor of the statistical analysis.No Mention of Stratification Beyond SDI. Though SDI is discussed, there's no clarity on whether analyses were stratified by sex, age group, or region, which would be important for understanding burden disparities. Explicitly state whether subgroup analyses were conducted, and if so, how these were handled statistically.Lack of Ethical Considerations Statement. While secondary data analysis typically does not require ethics approval, this should be explicitly stated for transparency. Add a sentence such as “This study used publicly available, de-identified secondary data and did not require ethical approval.”

**Results **

The term “smoking prevalence reached 40 per 100,000” seems inaccurate or misleading. Smoking prevalence is usually expressed as a percentage of the population, not per 100,000. Please revise to correct this. Similarly, DALYs and YLLs should be reported per 100,000 population, not as raw numbers or ambiguous “reached X per 100,000” without clear context.Break down the results more cleanly by subheading or paragraphing under the main result sections: E.g., under 3.1, consider sub-labels: “Regional Trends in Deaths,” “Patterns in DALYs,” etc. This enhances readability and helps readers locate specific data more easily.Ensure that all references to figures and tables are correctly numbered and labeled. “Figure 1.A” through “Figure 1.F” implies subfigures—these should be clearly shown in the actual figure. Ensure consistency between text and figures.You’ve done well to include 95% CIs and EAPC values in the table. However, where results are highlighted in the narrative (e.g., "a decline of -50.94%"), you might include the confidence interval briefly to strengthen the statistical transparency, especially for key figures.“...a decline of only -7.73%” consider avoiding “only” in scientific writing unless it's part of a contrastive point being emphasized."These findings highlight..." → consider beginning a new paragraph here to separate description of data from interpretation.Consistently use full terms on first mention in each subsection (e.g., "Years Lived with Disability (YLDs)").

**Discussion **

While you note the increased absolute Deaths and DALYs, there could be more exploration of why the smoking-attributable burden hasn't declined as sharply as overall smoking rates—e.g., delayed effects of smoking, demographic transitions.You mention comorbidities in older adults—can you tie this back to clinical practice implications, such as integrated care?The assertion that low SDI countries may underestimate burden due to data gaps or economic constraints is plausible but needs a supporting reference.“...all indicators (Deaths, DALYs, YLDs, and YLLs) , with...” → remove the space before the comma.

Reviewers' comments:

**Comments to the Author**

2. Has the statistical analysis been performed appropriately and rigorously? 

Reviewer #1: Yes

Reviewer #2: I Don't Know

3. Have the authors made all data underlying the findings in their manuscript fully available?

Reviewer #1: Yes

Reviewer #2: Yes

4. Is the manuscript presented in an intelligible fashion and written in standard English?

Reviewer #1: Yes

Reviewer #2: Yes

5. Review Comments to the Author

Reviewer #1: The manuscript presents an extensive examination of the relationship between smoking and the burden of rheumatoid arthritis (RA) across a wide range of populations and regions. Overall, the study is well-conducted and offers valuable insights using up-to-date data sources, such as the GBD 2021 and WHO databases. However, several improvements are necessary to further enhance clarity and presentation before publication.

Firstly, while the epidemiological metrics (e.g., Disability-Adjusted Life Years [DALYs], Years of Life Lost [YLLs], Years Lived with Disability [YLDs], Socio-Demographic Index [SDI], Uncertainty Interval [UI], Estimated Annual Percentage Change [EAPC], and Age-Standardized Rate [ASR]) are central to the analysis, their full forms should be clearly defined at first use. This change would ensure that readers from various disciplines can follow the arguments without confusion.

Secondly, the presentation of mathematical equations within the Methods section is currently embedded in the text and could be reformatted into stand-alone display equations with proper numbering and consistent variable notation. Enhanced clarity in the display of equations will aid in understanding the statistical approaches used and in verifying the methods.

Furthermore, the manuscript would benefit from a more consistent formatting of tables and figures. Table headers and figure captions should uniformly report units (e.g., per 100,000 population) and be detailed enough to stand alone. Special attention should be given to ensuring that all figures are of high resolution and that legends are clearly labeled.

Additionally, while the authors provide a detailed account of their methodological approach, incorporating a brief discussion of data quality—especially in low-income regions—and sensitivity analyses would strengthen the overall robustness of the study. Finally, minor language and typographical errors need correction, and administrative text should be streamlined to maintain focus on the science.

Reviewer #2: This manuscript provides a comprehensive picture of how smoking relates to rheumatoid arthritis outcomes worldwide. There were a few points of inconsistency throughout the manuscript, it is unclear if there may have been data missing that impacted the findings, and the methods could use more clarity.

1. Financial disclosure section says that “The author(s) received no specific funding for this work.” but there is a funding section of the paper which states “This work was supported by Scientific Research Fund of Hunan Provincial Education.” Please edit for accuracy and consistency.

2. Similarly, the data availability statement in entry fields at the start of the submission says, “Yes - all data are fully available without restriction” instead of including the data availability statement in the paper which includes the location of the data (lines 356-358, “The datasets analysed during the current study are available in the [Global Burden of”).

3. Abstract: Why is smoking referred to as an “external” risk factor? In the intro it is clear when you refer to genetic risk factors and environmental risk factors, but the use of “external” seems undefined.

4. Abstract is difficult to understand due to use of abbreviations without definitions throughout --- recommend to not include undefined abbreviations in the abstract.

5. Methods, “In GBD 2021, causes are categorized into four levels, ranging from Level 1 (communicable, maternal, neonatal, and nutritional diseases) to Level (latent tuberculosis infection).” Could you add in what these are levels of and how this information is important to the analysis in this paper?

6. Lines 172-173 need more information. They state, “In some Eastern European countries, smoking prevalence reached 40 per 100,000 while in sub-Saharan Africa, it was below 10 per 100,000.” These statistics appear incorrect, because smoking prevalence can be as high as 40% of the population, or 40 of every 100 people, in some countries. Either these statistics need to be updated or the sentence needs clarification to explain what these numbers are. Figure 1 similar shows in the keys that the statistics are per 100,000. At least for tobacco smoking and use, these numbers are actually likely per 100, not 100,000. I cannot speak to the other statistics because I am not as familiar with rheumatoid arthritis.

7. I’m having difficulty understanding table 1 DALYs. For example, Africa shows 6.22 for 1990 and 6.52 for 2021, but the % change is -41.78% --- this is confusing because there appears to be an increase from 1990 to 2021, not a decrease, and for example, the Americas has a similar change score below it, but the change from 10.91 in 1990 to 4.15 in 2021 appears to be a larger change and represents a decrease instead of an increase. It may be useful to define percentage change in the section titled, “2.2 ASR, EAPC and Percentage Change” --- the only information on percentage change in this section appears to be, “Additionally, this study used percentage change to represent the trend of the RA burden attributable to smoking in 2021 compared to 1990.” Other areas of Table 1 similarly show a decrease under percentage change but the 1990 to 2021 numbers appear to show an increase.

8. To happen during copy editing --- Figure 1 will need a higher pixel count. Very hard to read the graph legend, even after zooming in it is blurry. This applies to other figures as well.

9. Recommend checking the data in Figure 1, and assigning a specific color for countries where these is no data. For example, the figure shows that Russia has low smoking and tobacco use rates; however, other sources show that Russia has very high tobacco use and smoking rates: https://globalactiontoendsmoking.org/research/tobacco-around-the-world/russia/ . Potentially the data are missing for some countries and “0” was erroneously assigned. If it is found that data are missing from these figures with “0” erroneously assigned, I suggest that the authors check all data and statistics in the paper. Similarly, you note on lines 326-327, “First, data from low-income countries in the GBD database may be incomplete, potentially underestimating the actual burden.” Could you provide more information on how the data maybe incomplete, why you’re unable to identify if they are or are not incomplete, and how the lack of data may have impacted the statistics you report.

10. Line 231, “Among individuals aged 70-4” should likely be “70-74” ---recommend checking.

11. This paper could use a more thorough discussion of limitations and methods.

---

## [Author Response · Author response to Decision Letter 1]

20 Jun 2025

Journal Requirements:

and

Response:

Thank you for your kind reminder. In the revised manuscript submitted, we have carefully made adjustments and modifications to the formatting according to the requirements outlined in the two provided style template documents, to ensure compliance with PLOS ONE's standard formatting requirements.

2. We note that your Data Availability Statement is currently as follows: [All relevant data are within the manuscript and its Supporting Information files]

Response:

Thank you for the editor's kind reminder. To comply with PLOS ONE's data availability policy, we have prepared the minimal data set as follows: For each figure in our manuscript, we have created a dedicated folder containing all the underlying data points collated from the GBD 2021 and WHO website sources that were used to construct that figure. This packaged data has been uploaded to the submission system as Supporting Information.

Response:

Thank you for your reminder regarding the funding information. We have noted the discrepancy. The correct and consistent funding statement, which is accurately presented in our manuscript's "Funding" section and which we request be updated in the "Financial Disclosure" section of the online submission form, is as follows:

This work was supported by Scientific Research Fund of Hunan Provincial Education Department(grant number 23C0427, 23A0643), Hunan Provincial Natural Science Foundation of China (No.2021JJ40841), Research on education and teaching reform of Central South University (No. 2023jy087-3).

Response:

Thank you for your kind reminder. We have now updated the corresponding author's ORCID iD in the Editorial Manager submission system as requested.

“This work was supported by Scientific Research Fund of Hunan Provincial Education Department(grant number 23C0427, 23A0643), Hunan Provincial Natural Science Foundation of China (No.2021JJ40841), Research on education and teaching reform of Central South University (No. 2023jy087-3).”

Response:

Thank you for your kind reminder. We have updated the Funding Statement in the journal's online submission system. The updated funding information is as follows:

This work was supported by Scientific Research Fund of Hunan Provincial Education Department(grant number 23C0427, 23A0643), Hunan Provincial Natural Science Foundation of China (No.2021JJ40841), Research on education and teaching reform of Central South University (No. 2023jy087-3).

6. We note that Figure 1 in your submission contain [map/satellite] images which may be copyrighted. All PLOS content is published under the Creative Commons Attribution License (CC BY 4.0), which means that the manuscript, images, and Supporting Information files will be freely available online, and any third party is permitted to access, download, copy, distribute, and use these materials in any way, even commercially, with proper attribution. For these reasons, we cannot publish previously copyrighted maps or satellite images created using proprietary data, such as Google software (Google Maps, Street View, and Earth). For more information, see our copyright guidelines: http://journals.plos.org/plosone/s/licenses-and-copyright.

Response:

Thank you very much for your kind reminder. We paid close attention to copyright issues for all figures and potential conflicts of interest during our study design.

Regarding the concern about the copyright of Figure 1, we would like to clarify that the base map for this figure was generated using country boundary data (shapefile: ne_110m_admin_0_countries.shp) sourced from Natural Earth (naturalearthdata.com). Natural Earth data is in the public domain, which means it is free for all uses, including publication under a CC BY 4.0 license. The data for tobacco smoking rates, tobacco use rates, deaths, DALYs, YLDs, and YLLs, which are plotted onto this base map, were derived from our analysis of GBD 2021 and WHO data as described in the manuscript. Therefore, Figure 1 fully complies with the PLOS ONE CC BY 4.0 licensing requirements.

Furthermore, we have added a supplementary note to the caption of Figure 1:

Figure 1. Burden of RA attributable to smoking in 2021 across 204 countries and territories. (Base map data sourced from Natural Earth - public domain.)

Response:

Thank you for the reminder to carefully review our reference list. We have conducted a thorough review of all cited references in the manuscript, including those newly added during this revision process, to ensure their completeness, accuracy, and current publication status.

We confirm that, to the best of our knowledge and based on our checks, none of the references cited in our manuscript have been retracted.

Additional Editor Comments:

Introduction

1. While the authors mention that RA prevalence is rising globally, quantitative data or regional variability would enhance context. Consider including global or regional prevalence/incidence rates or burden metrics (e.g., DALYs or mortality). For example, in 2019, RA affected approximately X million people globally, contributing to Y DALYs and Z deaths (cite GBD source)."

Response:

Thank you for this valuable suggestion. We agree that providing quantitative data on the global burden of RA would significantly enhance the context of our introduction. We have revised the introduction to include specific global prevalence figures and the increase in its age-standardized prevalence rate for RA from the Global Burden of Disease (GBD) 2021 study, which is already cited as reference [3] in our manuscript.

The original sentence was:

Its prevalence is on the rise globally [3].

The revised sentence, now located on page 3, lines 36-38:

Globally, the prevalence of RA is on the rise, with an estimated 17.6 million people affected in 2020, and its age-standardized prevalence rate increasing by 14.1% since 1990 [3].

We believe this addition directly addresses your concern by providing concrete figures on the RA burden and its increasing trend, thus strengthening the rationale for our study on smoking-attributable RA.

2. Insufficient explanation of smoking’s pathophysiological role. Although smoking is cited as a major external risk factor, the introduction could briefly explain the mechanism by which smoking contributes to RA pathogenesis to help justify its inclusion. Add one or two sentences on how smoking induces citrullination, promotes autoantibody production, or triggers immune dysregulation relevant to RA.

Response:

Thank you for this insightful comment. We agree that a brief explanation of the pathophysiological link between smoking and RA would strengthen our introduction. We have now added a sentence to elaborate on this mechanism, supported by our existing reference [9] (Klareskog et al., 2011), as suggested.

The revised text, now located on page 3, lines 42-45:

Among these, smoking has been consistently identified as a significant external risk factor and is closely linked to the pathogenesis of RA [6-9]. Specifically, smoking is thought to promote RA development by inducing citrullination of proteins, particularly in the lungs of genetically susceptible individuals, which can lead to the generation of anti-citrullinated protein antibodies (ACPAs), key autoantibodies in RA [9].

3. Need to define the research gap more explicitly. The introduction could better articulate what specific gaps this study addresses. For example, is there a lack of global time-trend analyses? Have previous studies not linked smoking and RA in the context of FCTC implementation?

Response:

Thank you for pointing out the need to more explicitly define the research gap. We agree that this would strengthen the rationale for our study. We have revised the introduction to better articulate the specific gaps our study aims to address, particularly concerning the lack of comprehensive global time-trend analyses of smoking-attributable RA burden since the implementation of the WHO FCTC.

We have added the following sentences before stating our study's aim, now located on page 3, lines 54-59:

However, while the impact of smoking on RA is well-documented and the FCTC represents a global commitment to tobacco control, there is a paucity of comprehensive global analyses examining the long-term trends in smoking-attributable RA burden specifically since the FCTC's implementation. Understanding these trends is crucial for evaluating the real-world impact of tobacco control efforts on specific NCDs like RA and for guiding future public health strategies.

4. The final paragraph outlines the aim, but the sentence is lengthy and could be more concise and focused. For example, the objectives could be modified to “This study aims to assess global trends in the RA burden attributable to smoking since the implementation of the WHO FCTC, using data from GBD 2021 and WHO tobacco prevalence reports. Findings will inform health policy and tobacco control strategies aimed at reducing the RA burden.”

Response:

Thank you for this suggestion. We agree that our original statement of aims could be more concise and focused. We have revised the final paragraph of the introduction according to your an insightful suggestion.

The original sentence was:

This study aims to analyze and evaluate the epidemiological trends of RA burden at the global level since the implementation of the FCTC by utiliz

---

## [Editor Report · Decision Letter 1]

17 Jul 2025

Global Trends in Smoking-Attributable Rheumatoid Arthritis Burden: Insights from GBD 2021

PONE-D-25-01779R1

Dear Dr. Zhou,

We’re pleased to inform you that your manuscript has been judged scientifically suitable for publication and will be formally accepted for publication once it meets all outstanding technical requirements.

Kind regards,

Paul Obeng, MEd, MSc., M.Phil.

Academic Editor

PLOS ONE

---

## [Editor Report · Acceptance letter]

PONE-D-25-01779R1

PLOS ONE

Dear Dr. Zhou,

I'm pleased to inform you that your manuscript has been deemed suitable for publication in PLOS ONE. Congratulations! Your manuscript is now being handed over to our production team.

Kind regards,

on behalf of

Dr. Paul Obeng

Academic Editor

PLOS ONE